REGISTERED REPORT PROTOCOL

# HIV-associated nephropathy: Protocol and rationale for an exploratory genotype-phenotype study in a sub-Saharan African population

**Aminu Abba Yusuf** [1]☯*, **Baba Maiyaki Musa**[2,3‡], **Najibah Aliyu Galadanci**[4‡], **Musa Babashani**[2‡], **Aminu Zakari Mohammed**[5‡], **Donna J. Ingles**[6‡], **Agnes B. Fogo**[7☯], **C. William Wester**[6☯], **Muktar Hassan Aliyu**[6☯]

1 Department of Haematology and Blood Transfusion, Bayero University Kano/Aminu Kano Teaching Hospital, Kano, Nigeria, 2 Department of Medicine, Bayero University Kano/Aminu Kano Teaching Hospital, Kano, Nigeria, 3 Africa Center of Excellence for Population Health and Policy, 4 Department of Epidemiology, School of Public Health, University of Alabama at Birmingham, Birmingham, AL, United States of America, 5 Department of Histopathology, Bayero University Kano/Aminu Kano Teaching Hospital, Kano, Nigeria, 6 Vanderbilt Institute of Global Health (VIGH), Nashville, Tennessee, United States of America, 7 Department of Pathology, Microbiology and Immunology, Vanderbilt University Medical Center, Nashville, Tennessee, United States of America

☯ These authors contributed equally to this work.
‡ These authors also contributed equally to this work.
* aayusuf.hae@buk.edu.ng

This is a Registered Report and may have an associated publication; please check the article page on the journal site for any related articles.

## Abstract

### Background

HIV-positive persons of African descent are disproportionately affected by chronic kidney disease (CKD). Deterioration to end-stage kidney disease (ESKD) also occurs in this population at a higher frequency. There remains a lot to learn about the genetic susceptibility to CKD in HIV positive patients, and the pathophysiology of progression to ESKD.

### Objectives

We will conduct an exploratory genotype-phenotype study in HIV-positive persons with CKD in Aminu Kano Teaching Hospital, Nigeria, to determine blood-based differential gene expression biomarkers in different kidney risk groups according to the KDIGO 2012 criteria.

### Methods

We will consecutively screen 150 HIV-positive adults (≥18 years of age) attending the HIV clinic of Aminu Kano Teaching Hospital, Kano, Nigeria, for CKD based on proteinuria and elevation of estimated glomerular filtration rate. Among these, two separate groups of 16 eligible participants each (n = 32) will be selected in the four (4) KDIGO 2012 kidney risk categories. The groups will be matched for age, sex, viral suppression level and antiretroviral (ARV) regimen. In the first group (n = 16), we will determine differential gene expression markers in peripheral blood mononuclear cells using mRNA-sequencing (RNA-Seq). We

**Data Availability Statement:** All relevant data from this study will be made available upon study completion.

**Funding:** This work was supported by the Fogarty International Center at the U.S. National Institutes of Health (NIH) through the VECD Fogarty Global Health Fellowship Program (D43TW009337), URL: https://www.vumc.org/vecd/. The results and interpretations of this work are those of the authors and do not represent the official views of the NIH or the VECD Program. The funders had and will not have a role in study design, data collection and analysis, decision to publish, or preparation of the manuscript.

**Competing interests:** The authors have declared that no competing interests exist.

will validate the differential expression markers in the second group (n = 16) using reverse transcription quantitative polymerase chain reaction (RT-qPCR). Using a systems-based approach, we will construct, visualize and analyze gene-gene interaction networks to determine the potential biological roles of identified differential expression markers based on published literature and publicly available databases.

## Results

Our exploratory study will provide valuable information on the potential roles of differential expression biomarkers in the pathophysiology of HIV-associated kidney disease by identifying novel biomarkers in different risk categories of CKD in a sub-Saharan African population. The results of this study will provide the basis for population-based genome-wide association studies to guide future personalized medicine approaches.

## Conclusion

Validated biomarkers can be potential targets for the development of stage-specific therapeutic interventions, an essential paradigm in precision medicine.

## Introduction

Human immunodeficiency virus (HIV) positive persons are at increased risk of kidney disease. HIV-associated nephropathy (HIVAN) disproportionately affects persons of African descent, with a prevalence of 3–12% and a more severe clinical presentation than observed in non-African populations [1, 2]. HIVAN is characterized by a collapsing focal segmental glomerulosclerosis lesion, but other forms of kidney linked to HIV have also been described [3, 4]. Clinically, HIVAN presents as proteinuria and rapidly progresses to ESKD [5]. Alterations in the host gene expression in key cellular pathways are implicated in kidney-related pathology in HIV disease [6–8]. Considerable diversity in genetic susceptibility exists among patients, and risk allele variants in the apolipoprotein-L1 (*APOL1*) gene strongly increase HIVAN risk in HIV-positive blacks [9]. Deregulation in pathways involved in cell cycle, inflammation and apoptosis ultimately results in this unique histopathological disease. These events are directly linked to the level of glomerular injury, proteinuria and decline in estimated glomerular filtration rate (eGFR) [10].

Many questions remain unanswered concerning the pathophysiology of kidney disease in HIV [1]. Although *APOL1* variants probably account for a majority of the increased susceptibility to focal segmental glomerulosclerosis and HIVAN, they fail to explain a significant portion of the disease. In one study, *APOL1* risk variant alleles did not account for all susceptibility to HIVAN and poorly predicted disease characteristics, including progression to ESKD or death [11]. Much of the genetic susceptibility in 30% of individuals without the *APOL1* high-risk genotype (i.e. those with 0 or 1 *APOL1* risk alleles) is yet to be explored. Further, although *APOL1* variants are postulated to increase susceptibility to kidney disease by effects on podocytes and their cytoskeleton, the exact mechanism through which they cause kidney disease is not proven. Additionally, whether *APOL1* is an initiator of HIVAN or a progression factor or both is yet to be ascertained [4].

Most studies of kidney disease in HIV are focused on genetic risk factors for developing kidney disease [9, 10]; few studies address the genetic risks associated with progression to

ESKD, and current therapies do not target genetic risks as linked to distinct HIV-related pathways [4]. Kidney biopsies are invasive and not indicated in all patients. Whether phenotypically determined kidney risk groups based on eGFR and albuminuria have differential gene expression signatures in HIV-positive persons living in sub-Saharan Africa has not been previously explored. To address this gap, we propose a study to explore biomarkers differentially expressed in different risk categories of kidney disease in HIV-positive patients. We hypothesized that blood-based host gene expression biomarkers could define disease progression through stages that correspond to distinct phenotypic categories based on eGFR and albuminuria. Identification of these markers and their roles in the pathology of kidney disease in HIV is critical to elucidating the molecular basis of progression to ESKD, and uncovering potential therapeutic targets for precision medicine.

## Methods

### Setting and study population

The study will be conducted at Aminu Kano Teaching Hospital (AKTH) located in Kano, northwestern Nigeria. Kano is the most populated state in Nigeria. The majority of eligible participants are expected to be ethnic blacks who are predominantly Hausa speaking, although other ethnicities are also likely to be represented. Participants will be recruited during routine visits to the outpatient antiretroviral clinics of the hospital HIV clinic. The HIV clinic, named S.S. Wali Center, is a U.S. President's Emergency Plan for AIDS Relief (PEPFAR)-funded clinic that operates five days a week. The large outpatient clinic has a total enrolment of more than 10,000 HIV-positive patients with an average 200 daily visits.

### Study design

The study is designed as an exploratory cross-sectional genotype-phenotype study exploring gene expression biomarkers at different clinical stages of kidney disease in HIV-positive persons who are on antiretroviral (ARV) therapy. A total of 150 HIV-positive adults will initially be screened for proteinuria and reduced eGFR as markers of kidney disease. From this group, we will select 32 participants at different clinical stages of kidney disease corresponding to the four kidney risk groups based on the KDIGO 2012 Guideline [12] for genetic testing. Selected participants will be comprised of both virologically- suppressed and non-virologically suppressed ART-treated adults to ensure adequate representation of the different clinical scenarios in the study.

### Specific aims

The specific aims of the study are:

1. To identify individuals with HIV-associated kidney disease based on phenotype and clinical staging of kidney disease that correspond to risk groups based on similar relative risks of kidney outcomes, as determined by proteinuria and eGFR staging.

2. To explore differential gene expression markers at different clinical stages of HIV-associated kidney disease in PBMC from patients pre-selected by proteinuria and eGFR level in Aim 1, using mRNA-sequencing and RT-qPCR techniques.

3. To identify the biological roles of differentially expressed markers in different kidney risk categories through a post hoc literature search and construction and analysis of gene-gene interaction networks using Cytoscape.

## Sample size and participant selection

We plan to use non-probability convenience sampling method to screen 150 consecutive patients for evidence of kidney disease using a combination of proteinuria and eGFR as markers, and according to the KDIGO 2012 criteria. Of these, we will select 32 eligible participants using purposeful sampling for this exploratory genotype-phenotype study.

## Participant recruitment

Participants will be recruited from the adult HIV clinic of AKTH during routine follow-up visits. Basic demographic information such as age, sex and self-declared ethnicity will be collected using a structured, interviewer-administered questionnaire.

## Study population

**Inclusion criteria.**   The inclusion criteria for this study include: 1. Age $\geq$ 18 years, 2. HIV-positive, 3. Enrolled as a patient in the AKTH HIV clinic 4. On antiretroviral therapy (ART). Diagnosis of HIV infection will be per local protocol using rapid diagnostic kits (Stat-pack or UniGold) for screening and HIV-1 enzyme-linked immunosorbent assay (ELISA) for confirmation. Moderately increased albuminuria will be defined as a mean uACR of 30–300 mg/g. CKD will be defined as the presence of proteinuria or an eGFR of $< 60$ ml/min/1.73m$^2$. Consistent with the KDIGO 2012 definition of CKD, uACR and eGFR measurements will be repeated 3months after the initial testing to ensure these findings are persistent before selection. Selected participants for the RNA-Seq and RT-qPCR validation ($n$ = 16 each) will be matched for age range (within five years interval), sex, virologic suppression status and ARV regimen to minimize potential confounding.

**Exclusion criteria.**   The following criteria will be used to exclude potential participants: 1. Presence of bacteriuria defined (based on urine culture), 2. Positive nitrites ($\geq$1+) on urine dipstick, 3. Pregnancy 4. Any known malignancy and 5. Diagnosis of diabetes mellitus.

## Study procedures

**Research ethics.**   *Institutional ethics approval.* Ethics clearance to conduct the study was obtained from the AKTH Research Ethics Committee (REC) (approval number: NHREC/21/08/2008/AKTH/EC/2444). All study procedures will be conducted in accordance with the principles of the Helsinki Declaration.

**Informed consent.**   Informed written consent will be obtained from potential participants before enrolment into the study. A trained study team member will explain to each potential participant the nature of the study, its purpose, the procedures involved, the expected duration, the potential risks and benefits involved, and any discomfort it may entail. The recruitment script will be made available in both English and Hausa language to aid comprehension. Participants will be informed about the voluntary nature of participation and their right to withdraw from participation at any time without affecting their care at AKTH. Participants will also be informed about the reimbursement they will receive for participation in this study (to compensate them for transport expenses and for time away from work/home while participating in this study).

A separate informed consent will be obtained from participants selected for genetic testing (RNA-Seq and RT-qPCR) in which specific details regarding this testing will be explained to them.

Beneficence. Study participants who are found to have abnormal proteinuria or eGFR will be referred to the Nephrology Clinic at AKTH for further evaluation and treatment according to the standard of care at AKTH.

## Study procedures/recruitment

For Aim 1, we will sequentially approach and recruit eligible HIV-positive adults attending the HIV clinic at AKTH and screen them for proteinuria using a Combi-9 dipstick urinalysis strip on spot urine specimen. Protein dipstick-negative patients will be tested for micro-albuminuria through measurements of urinary albumin-creatinine ratio (uACR) on a spot urine specimen. Patients will be classified as normal albumin excretion (uACR <30mg/g), moderately increased albuminuria (uACR = 30-299mg/g) or severely increased albuminuria (uACR ≥300mg/g). Individuals in either of the latter two groups will be classified as having kidney disease. Serum cystatin C level will be measured to enable eGFR calculations.

Using the composite of albumin excretion and serum cystatin C-based eGFR calculations, screened participants will be grouped into four risk groups based on similar relative risks of kidney outcomes according to the KDIGO (2012) guidelines shown below [12]. See Fig 1.

i. Group I: No CKD

ii. Group II: Moderately increased risk

iii. Group III: High risk

iv. Group IV: Severely increased risk

For Aim 2, we will select a total of 32 participants from the initial 150 screened participants from each of the four risk categories above for genetic testing. Per each of group i-iv above, four participants (*n* = 16) will be selected for RNA-Seq differential expression analysis (as

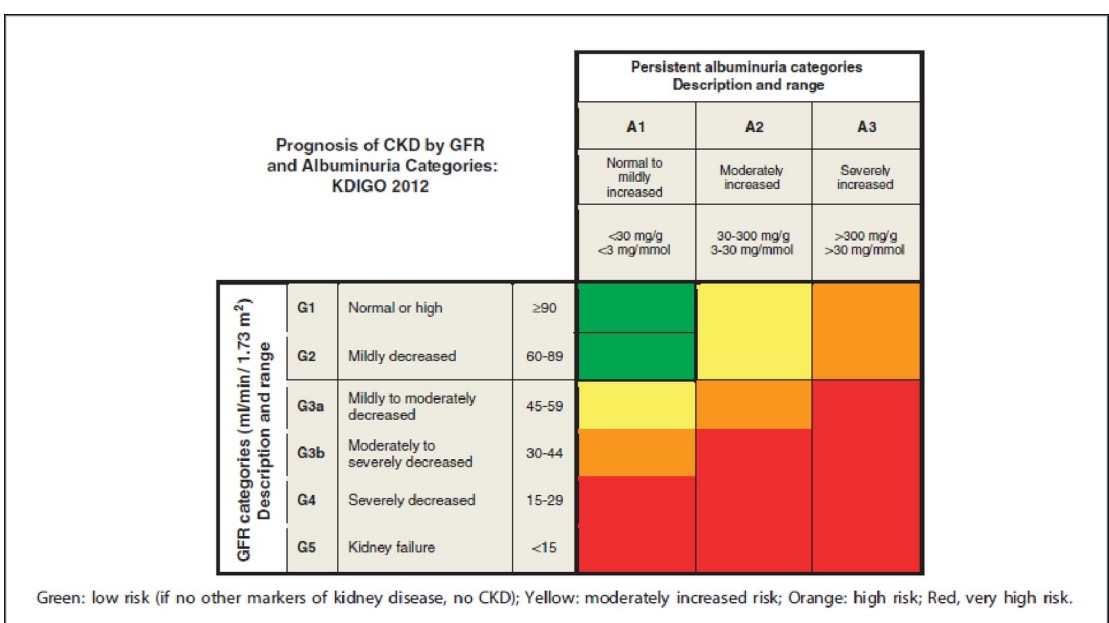

**Fig 1. Prognosis in CKD based on albuminuria and eGFR category.** (Image adapted from Ref. No. 9, with permission from the KDIGO).

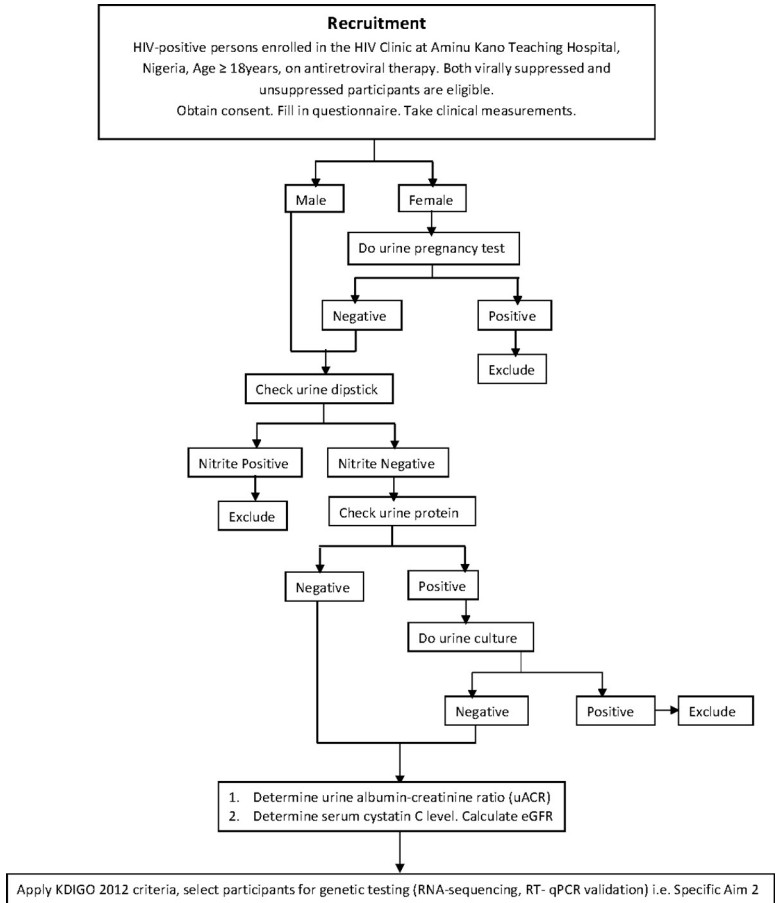

**Fig 2. Participant recruitment algorithm and study procedures.**

'Discovery Cohort') and 16 participants for RT-qPCR validation (as 'Validation Cohort'). See Fig 2.

For Aim 3, we will search literature in publicly available repositories (PubMed and Expression Atlas) for potential roles of identified markers in the pathophysiology of kidney disease in HIV. Using a systems approach, we will use Cytoscape to construct, visualize and analyze potential gene-gene interaction networks to depict the molecular interactions between the differentially expressed genes based on known interactions published in the literature.

## Data collection

**Laboratory analyses.** Urinary albumin concentration will be determined using the immunoturbidometric method on a Hemocue machine, while urinary creatinine will be assayed using the Jaffe kinetic method. Albumin-creatinine ratios will then be calculated and expressed in mg/g unit to assess albuminuria. Venous blood (5ml) will be obtained by venipuncture in a gel clot-activator specimen bottle. Serum cystatin C will be assayed using the Roche Integra 800 system (Roche Diagnostics Indianapolis, IN).

**Genetic specimen collection and analysis.** Four (4) representative samples from each of the four KDIGO 2012 kidney risk groups ($n = 16$) will be selected for genetic analysis using mRNA-Seq as the 'Discovery Cohort'. Venous blood (10ml) will be collected from a forearm vein by an aseptic venipuncture into ethylene diamine tetraacetic acid (EDTA)-anticoagulated

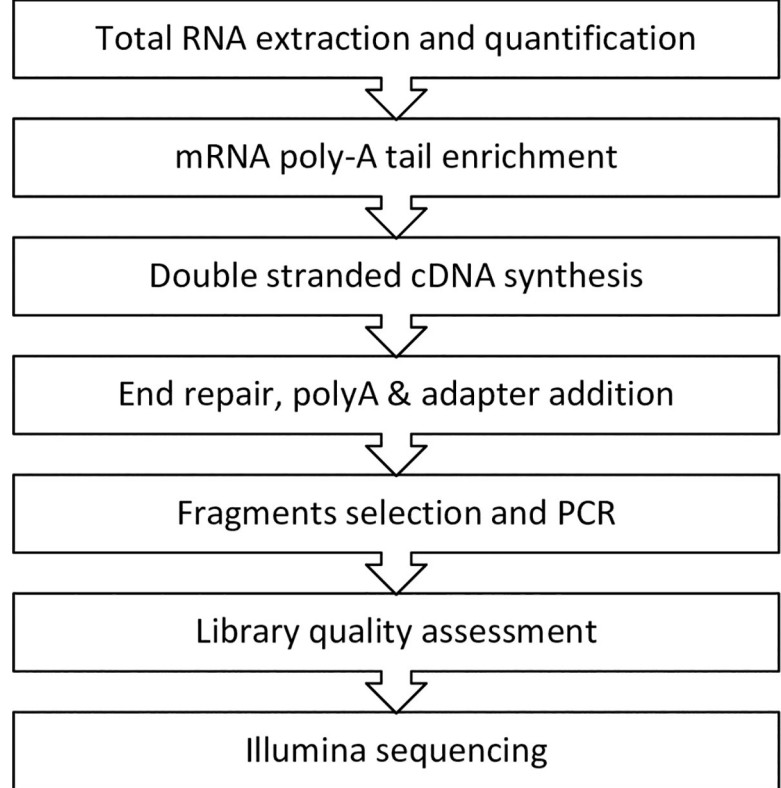

**Fig 3. Experimental workflow for RNA Illumina sequencing.**

specimen tubes. These samples will be stored at room temperature and processed within two hours of collection. We will initially isolate PBMC using the Ficoll-Paque differential gradient technique (Amershem Biosciences, Sweden) from which we will extract total RNA using the RNAeasy Plus Mini kit (Qiagen, Germany), following the manufacturer's instructions. Extracted RNA will be subjected to quality control (QC) to ensure acceptable RNA quality before proceeding to the next step. Additional isolation and poly-A tail enrichment will be performed to isolate mRNA using oligo(dT) beads. mRNA-Seq will be performed on a MiSeq sequencing platform (Illumina) to identify differentially expressed (DE) genes per manufacturer instructions. Fig 3 shows the experimental workflow for the RNA-Seq experiment.

To validate DE markers found using RNA-Seq, we will design specific primers for the identified DE genes, and conduct RT-qPCR for differential expression as a validation step in a separate cohort of 16 participants (i.e. the 'Validation Cohort') selected from the initially screened 150 participants using the same KDIGO 2012 criteria. The 'Discovery Cohort' and the 'Validation Cohort' will be matched for age range (5 years), sex, viral suppression level, and ARV regimen. (Fig 4).

## Plan for data and statistical analyses

**RNA-Seq and RT-qPCR data analysis.** Analyses of the raw RNA-Seq data will follow the standard workflow, which includes mapping, transcript assembly and quantification, splice variants and junctions, normalization, differential expression analysis, gene ontology (GO) enrichment analysis and biological interpretation. Newly identified sequence reads will be aligned to a reference genome (*Homo sapiens*). Following alignment, many variations may be

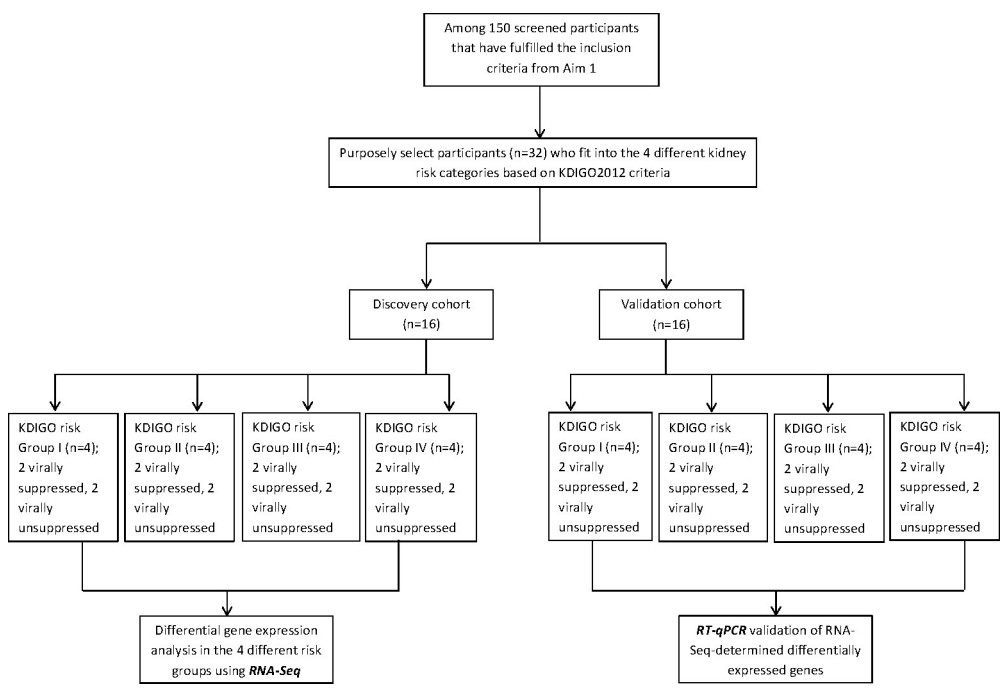

**Fig 4. Participants selection for genetic testing.**

identified such as single nucleotide polymorphisms or insertion-deletion. We will use R software (v2009-2014, Bioconductor) to compute mean expression levels for identified genes across the different samples. These expression levels will be compared using analysis of variance (ANOVA) for statistical significance. Principal component analyses (PCA) and hierarchical cluster analyses will be conducted to reduce the dimension of large data sets and to explore sample classes arising naturally based on the expression profile.

Five hundred (500) genes with the largest coefficient of variation based on fragment per kilobase of exon model count abundance estimations will be included in the PCA and will be represented in a PCA plot to show sample clustering. If the biological differences between the samples are pronounced, the plot will describe the primary components of the variation in the data and will lead to separation of samples in different regions of the PCA plot corresponding to their biology. However, if other factors (e.g. sample quality) introduce more variation in the data, the samples may not cluster according to the biology. The HCA will be presented in a heat map plot to demonstrate sample and gene clustering based on normalized abundance.

Differential expression analysis will be performed using the DESeq method [13]. The resulting $p$-values will be adjusted using Benjamini and Hochberg's approach for controlling the false discovery rates (FDR). Those genes that have a twofold difference (the absolute value of log2 ratio $\geq 1$) between two stages and FDR less than 0.01 will be identified as differentially expressed genes. Similar data analysis will be employed for differential expression experiments for RT-qPCR.

We will conduct a post hoc literature search in publicly available databases to determine the potential biological roles of the validated DEGs as related to the development or progression of kidney disease in HIV. Gene function will be annotated based on the Kyoto Encyclopedia of Genes and Genomes (KEGG) and gene ontology (GO) databases. Using a systems biology approach, we will use Cytoscape software (v3.8.0, 2018) [14] to construct and visualize potential gene-gene interaction networks using the vizMap and Agilent Literature Search plug-ins

**Table 1. Proposed study timelines.**

| ACTIVITIES | MONTHS | | | | | | | | | | | |
|---|---|---|---|---|---|---|---|---|---|---|---|---|
| | 1 | 2 | 3 | 4 | 5 | 6 | 7 | 8 | 9 | 10 | 11 | 12 |
| **Protocol development** | X | | | | | | | | | | | |
| **Ethics approval** | X | X | | | | | | | | | | |
| **Hire and train study staff** | X | X | | | | | | | | | | |
| **Data collection** | | | | | | | | | | | | |
| • Informed consent for **Aim 1** | | | X | X | X | X | X | | | | | |
| • Screening & recruitment for **Aim 1** –Questionnaire, uACR and eGFR measurements | | | | X | X | X | X | X | | | | |
| • Informed consent for genetic analysis (**Aim 2**) | | | | | | | X | X | | | | |
| • Sample collection, processing for genetic analysis (**Aim 2**) | | | | | | | X | X | X | | | |
| • Genetic testing–mRNA-Sequencing, qPCR validation (**Aim 2**) | | | | | | | | | X | X | X | |
| **Data management and QC** | | | | | | | X | X | X | X | X | |
| **Data analysis (Aim 3)** | | | | | | | X | X | X | X | X | |
| **Dissemination of study findings** | | | | | | | | | | | X | X |

*uACR: urinary albumin-creatinine ratio; eGFR: estimated glomerular filtration rate; QC: quality control.

based on the RNA-Seq differential expression data, to evaluate potential interactions among the identified DE markers. We will present all data in conformity with the minimum information about a genome sequence specification [15] and minimum information about quantitative PCR experiment guidelines [16]. We will select the most promising protein product(s) of the genes identified as being differentially expressed and based on their potential biological or pathophysiological roles in kidney disease in HIV. We will use the ELISA method to quantify serum protein levels across the KDIGO groups to determine if these correlate with gene expression patterns and/or the severity of kidney disease. Table 1 shows the proposed timeline of study activities.

## Discussion

CKD remains a major problem among HIV-positive populations and disproportionately affects individuals of African descent [1]. The largest number of HIV-positive persons in the world resides in sub-Saharan Africa, constituting 65% of the 37.9million people living with HIV/AIDS globally [17]. This high burden of HIV disease, the diverse genetic backgrounds of the population, and a multitude of environmental influences make sub-Saharan Africa a significant region in which to conduct genotype-phenotype studies in HIV infection.

In this exploratory study, we will assess whether there are blood-based gene expression biomarkers that correlate with phenotypically determined kidney risk groups based on a composite of two key markers of CKD (urine albumin excretion and eGFR). The risk for morbidity and mortality in CKD has been validated in large cohorts of CKD patients [18]. The kidney risk groups forming the phenotypic groups in this study were adopted from KDIGO 2012 guidelines [12], which are widely used in clinical settings and have the utility of assigning risk for morbidity and mortality outcomes in diverse patient populations with CKD from various etiologies, including HIV infection. This risk categorization was chosen because it has direct clinical relevance, as it is derived based on eGFR/albuminuria categories with similar relative risks of CKD outcomes [12]. As nephropathy can also occur in patients with undetectable HIV-1 RNA levels [19], we will recruit both virally suppressed and unsuppressed patients.

We chose to use PBMC for our gene expression studies, as they represent the main targets for direct HIV infection, and studies have shown PBMC gene expression to be deregulated by

HIV infection [20–23]. In addition, PBMC are easily obtained from the peripheral blood, making them suitable for biomarker assessments. Further, deregulation of gene expression or HIV viral load in PBMC mirrors HIV disease progression [24, 25] or the propensity for end-organ damage in HIV infection, with significant prognostic implications [26].

RNA-Seq offers a unique opportunity in functional genomics over other techniques such as qPCR and DNA microarray because it is a high throughput technology and ideally suited for exploratory studies. Additionally, RNA-Seq offers the prospect of discovering novel transcripts such as alternative splicing and novel SNP variants, in addition to known insertion-deletion variants. Integrating functional genomics (RNA-Seq) with a systems-based approach will also improve the prospect of uncovering potential key molecular interactions important in understanding the pathogenesis of kidney disease in HIV infection. We chose a 'validation cohort' separate from the 'discovery cohort' to provide external validity and hence improve the generalizability of the potential research findings.

## Conclusion

Our exploratory study will provide valuable information on the potential roles of differential expression biomarkers in the pathophysiology of HIV-associated kidney disease by identifying novel biomarkers in different risk categories of CKD in a sub-Saharan African population. The project could also provide the basis for population-based genome-wide association studies. Results from this study could provide the preliminary data necessary for future studies to assess changes in putative markers over time in a given patient and assess if they predict or correlate with CKD progression. Validated biomarkers can also be potential targets for the development of stage-specific therapeutic interventions, an essential paradigm in precision medicine.

## Acknowledgments

The authors wish to thank the Management of AKTH for agreeing to provide the facility support to conduct this research, and the Management of Bayero University Kano and AKTH for agreeing to grant the protected time to conduct this research.

## Author Contributions

**Conceptualization:** Aminu Abba Yusuf, Baba Maiyaki Musa, Musa Babashani, Agnes B. Fogo, C. William Wester, Muktar Hassan Aliyu.

**Funding acquisition:** Aminu Abba Yusuf, Donna J. Ingles, C. William Wester, Muktar Hassan Aliyu.

**Methodology:** Aminu Abba Yusuf, Najibah Aliyu Galadanci, Aminu Zakari Mohammed, Donna J. Ingles, Agnes B. Fogo, C. William Wester, Muktar Hassan Aliyu.

**Project administration:** Aminu Abba Yusuf, Donna J. Ingles, Muktar Hassan Aliyu.

**Supervision:** Aminu Abba Yusuf, Baba Maiyaki Musa, Najibah Aliyu Galadanci, Musa Babashani, Aminu Zakari Mohammed, Agnes B. Fogo, C. William Wester, Muktar Hassan Aliyu.

**Validation:** Aminu Abba Yusuf.

**Writing – original draft:** Aminu Abba Yusuf, Baba Maiyaki Musa, Najibah Aliyu Galadanci, Musa Babashani, Aminu Zakari Mohammed, Agnes B. Fogo, C. William Wester, Muktar Hassan Aliyu.

**Writing – review & editing:** Aminu Abba Yusuf, Baba Maiyaki Musa, Najibah Aliyu Galadanci, Musa Babashani, Aminu Zakari Mohammed, Donna J. Ingles, Agnes B. Fogo, C. William Wester, Muktar Hassan Aliyu.

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
