## [Decision Letter · Decision Letter 0]

28 Oct 2020

PONE-D-20-21492

HIV-associated nephropathy: Protocol and rationale for an exploratory genotype-phenotype study in a sub-Saharan African population

PLOS ONE

Dear Dr. Yusuf,

Thank you for submitting your manuscript to PLOS ONE. After careful consideration, we feel that it has merit but does not fully meet PLOS ONE’s publication criteria as it currently stands. Therefore, we invite you to submit a revised version of the manuscript that addresses the points raised during the review process.

Please respond to the reviewers' comments, especially those concerning sample size.

We look forward to receiving your revised manuscript.

Kind regards,

Nancy Beam, PhD

Staff Editor

PLOS ONE

Journal Requirements:

2. Please address the following:

- Please refer to any sample size calculations performed prior to participant recruitment. If these were not performed please justify the reasons. Please refer to our statistical reporting guidelines for assistance (https://journals.plos.org/plosone/s/submission-guidelines.#loc-statistical-reporting).

- Please include additional information regarding the survey or questionnaire that will be used in the study and ensure that you have provided sufficient details that others could replicate the analyses. For instance, if you developed a questionnaire as part of this study and it is not under a copyright more restrictive than CC-BY, please include a copy, in both the original language and English, as Supporting Information.

4. We note you have included a table to which you do not refer in the text of your manuscript. Please ensure that you refer to Table 2 in your text; if accepted, production will need this reference to link the reader to the Table.

Reviewers' comments:

Reviewer's Responses to Questions

**Comments to the Author**

1. Does the manuscript provide a valid rationale for the proposed study, with clearly identified and justified research questions?

Reviewer #1: Yes

Reviewer #2: Yes

2. Is the protocol technically sound and planned in a manner that will lead to a meaningful outcome and allow testing the stated hypotheses?

Reviewer #1: Yes

Reviewer #2: Yes

3. Is the methodology feasible and described in sufficient detail to allow the work to be replicable?

Reviewer #1: Yes

Reviewer #2: Yes

4. Have the authors described where all data underlying the findings will be made available when the study is complete?

Reviewer #1: Yes

Reviewer #2: Yes

5. Is the manuscript presented in an intelligible fashion and written in standard English?

Reviewer #1: Yes

Reviewer #2: Yes

6. Review Comments to the Author

You may also provide optional suggestions and comments to authors that they might find helpful in planning their study.

Reviewer #1: The report protocol “HIV-associated nephropathy: Protocol and rationale for an exploratory genotype-phenotype study in a sub-Saharan African population” by Aminu Abba Yusuf et al., describes an exploratory study which intend to associate the potential differential expression biomarkers of HIV infection with chronic kidney disease using the next generation sequencing technique, RNASeq, and subsequently validate the altered gene expressions using real-time PCR technique. The authors described the rationale of the study well and I believe that the study has an important impact, which potentially help in understanding the physiological development of chronic kidney disease in people living with HIV. However, there are few concerns that I wish to clarify with the authors regarding the study design, which listed as below:

1. The authors mentioned that 150 of HIV-positive adults will be screened for Chronic Kidney Disease, and 32 individuals from the same cohort will be selected for the downstream analysis. Of the 32 individuals, 16 of them will be grouped into the discovery group, where sample of these subjects will be used for RNASeq analysis, while the other 16 samples will be grouped into the validation group, which their RNA samples will be subjected to PCR analysis. I wish to ask the authors, is there a reason why the subjects are separated in such a way? Why can’t the validation to be performed on the same subjects of the discovery cohort?

2. 16 patients from the discovery group and validation group will be further assigned equally into non-ckd, mild-ckd, moderate-ckd, and severe-ckd where in each group would only consist of four patients. Would such a low sample size in each group able to achieve a valid statistical significant? I do think that the sample size is rather small. The authors should consider to increase the number of samples in order to achieve a valid statistical difference.

3. Would there be any plan of exploring the proteins that expressed by the deferential expressed gene in plasma samples? I think this would have added value to the finding as circulating plasma proteins may be a better predictor for the disease compared to RNA sample.

Reviewer #2: This is a well written manuscript. The study is important and an area where further exploratory work needs to be done

Just a few minor comments

Introduction

1. Line 72 - in addition to reference 3 [ would suggest reference 5 as well

2. Please check reference 6 : it does not seem appropriate to the information for this reference. This reference is not a genetic based study

3. Line 84 : this last section needs a reference

Specific aims

line 119 : need to remove the word "of"

Inclusion criteria

- It is not clear whether all patients need to be on ARVs or not and not clear in methods that patients do not need to be virally suppressed. It is mentioned later on in the text regarding viral suppression however this should be clearly stated in the methods. I needed to go back and forth to review and try and figure this out and would assist if it was more specifically mentioned in the methods. Plus line 146 ARV has not been written out in full earlier in the manuscript

Exclusion criteria

- Not sure whether necessary to clearly state that < 18 years is an exclusion criteria  it is in the inclusion

Figure 2 : This diagram does not stand alone : It would help if you stated in the recruitment ...the population you are recruiting

Figure 4 : It would also assist to state the population you are working with and whether they are on ART or not * This is not clear in the methods

These are very minor additions however I feel that they would assist the reader with clarity

7. PLOS authors have the option to publish the peer review history of their article (what does this mean?). If published, this will include your full peer review and any attached files.

Reviewer #1: No

Reviewer #2: **Yes: **Prof Nicola Wearne

---

## [Author Response · Author response to Decision Letter 0]

22 Nov 2020

Thank you for the review of our submission to PLOS One. We have now provided a point-by-point response to the reviewer comments and a revised version of our article with changes highlighted in color font. A second “clean” copy of the revised submission is also included. 

Reviewer #1

The authors mentioned that 150 of HIV-positive adults will be screened for Chronic Kidney Disease, and 32 individuals from the same cohort will be selected for the downstream analysis. Of the 32 individuals, 16 of them will be grouped into the discovery group, where sample of these subjects will be used for RNASeq analysis, while the other 16 samples will be grouped into the validation group, which their RNA samples will be subjected to PCR analysis. I wish to ask the authors, is there a reason why the subjects are separated in such a way? Why can’t the validation to be performed on the same subjects of the discovery cohort?

Response: We chose to validate on a separate cohort in order to provide external validity to our findings and thereby improve the generalizability of the research findings. We have added this clarification to the discussion section of the manuscript (Page 14, lines 298-299).

2. 16 patients from the discovery group and validation group will be further assigned equally into non-ckd, mild-ckd, moderate-ckd, and severe-ckd where in each group would only consist of four patients. Would such a low sample size in each group able to achieve a valid statistical significant? I do think that the sample size is rather small. The authors should consider to increase the number of samples in order to achieve a valid statistical difference.

Response: The sample size for this preliminary exploratory study was a convenience pilot sample and governed by available funding. We will revisit the possibility of increasing the sample size and any decisions will be based on the prevailing situation.

3. Would there be any plan of exploring the proteins that expressed by the deferential expressed gene in plasma samples? I think this would have added value to the finding as circulating plasma proteins may be a better predictor for the disease compared to RNA sample.

Response: Initially, we did not plan to explore protein expression levels for this phase of the project, as we currently do not know which genes will be differentially expressed among the groups. We planned protein expression studies and validation in larger populations after this exploratory study, and that will be the next phase of the study. However, based on the reviewer’s suggestion, we will select the most promising protein products of the genes we’ve identified as being differentially expressed and based on their potential biological or pathophysiological roles in kidney disease in HIV. We will use the ELISA method to quantify serum protein levels across the KDIGO groups to determine if these correlate with gene expression patterns and/or the severity of kidney disease. (Please see Page 12, lines 257-261).

Reviewer #2: This is a well written manuscript. The study is important and an area where further exploratory work needs to be done

Response: Thank you for the comment. 

Just a few minor comments

Introduction

1. Line 72 - in addition to reference 3 [ would suggest reference 5 as well

Response: Both Reference 5 and Reference 3 have been added to the Introduction as recommended. References 3 and 5 in the original manuscript are now references 3 and 4 in the revised copy as rearranged (Page 4, line 73).

2. Please check reference 6: it does not seem appropriate to the information for this reference. This reference is not a genetic based study

Response: Reference 6 was a review article that dealt (in part) with the topic. However, we have now replaced it with 3 original research article references (Refs #6-8) that described the information in our text. Thank you. (See reference section please)

3. Line 84: this last section needs a reference

Response: A reference has now been added to this section. Please see the new reference #4, now on line 89, Page 4.

Specific aims

line 119: need to remove the word "of"

Response: “Of’ has been now removed. Thank you. 

Inclusion criteria

- It is not clear whether all patients need to be on ARVs or not and not clear in methods that patients do not need to be virally suppressed. It is mentioned later on in the text regarding viral suppression however this should be clearly stated in the methods. I needed to go back and forth to review and try and figure this out and would assist if it was more specifically mentioned in the methods. Plus line 146 ARV has not been written out in full earlier in the manuscript

Response: Viral suppression status of participants now stated in the methods section. ARV now fully spelled out the first time it was used in the manuscript, in Page 5, Line 114, as requested. Thank you.

Exclusion criteria

- Not sure whether necessary to clearly state that < 18 years is an exclusion criterion  it is in the inclusion

Response: Thank you. We have removed “Age < 18 years” in the exclusion criteria text since it has been mentioned in the inclusion criteria.

Figure 2 : This diagram does not stand alone : It would help if you stated in the recruitment ...the population you are recruiting

Response: Information on the recruitment population has now been added to the diagram to make it stand alone. (see Fig 2 please)

Figure 4 : It would also assist to state the population you are working with and whether they are on ART or not * This is not clear in the methods

Response: Information on the recruitment population and their ART status have now been added to this diagram to make it stand alone. (see Fig 4 please). This has also been clarified in the methods. (Page 6, line 118)

These are very minor additions however I feel that they would assist the reader with clarity

7. PLOS authors have the option to publish the peer review history of their article (what does this mean?). If published, this will include your full peer review and any attached files.

Response: Yes, we agree to have the peer review history to be published.

Once again, we thank you very much for the consideration of our manuscript and for the reviews. We look forward to a favourable decision and publication of our manuscript in your esteemed Journal.

Sincerely,

Aminu Abba Yusuf

Corresponding author

On behalf of the authors.

---

## [Decision Letter · Decision Letter 1]

22 Mar 2021

HIV-associated nephropathy: Protocol and rationale for an exploratory genotype-phenotype study in a sub-Saharan African population

PONE-D-20-21492R1

Dear Dr. Yusuf,

We’re pleased to inform you that your manuscript has been judged scientifically suitable for publication and will be formally accepted for publication once it meets all outstanding technical requirements.

Kind regards,

Nicole Endlich, Prof

Academic Editor

PLOS ONE

Additional Editor Comments (optional):

Reviewers' comments:

Reviewer's Responses to Questions

**Comments to the Author**

1. Does the manuscript provide a valid rationale for the proposed study, with clearly identified and justified research questions?

Reviewer #1: Yes

Reviewer #2: Yes

Reviewer #3: Yes

2. Is the protocol technically sound and planned in a manner that will lead to a meaningful outcome and allow testing the stated hypotheses?

Reviewer #1: Yes

Reviewer #2: Yes

Reviewer #3: Partly

3. Is the methodology feasible and described in sufficient detail to allow the work to be replicable?

Reviewer #1: Yes

Reviewer #2: Yes

Reviewer #3: Yes

4. Have the authors described where all data underlying the findings will be made available when the study is complete?

Reviewer #1: Yes

Reviewer #2: Yes

Reviewer #3: Yes

5. Is the manuscript presented in an intelligible fashion and written in standard English?

Reviewer #1: Yes

Reviewer #2: Yes

Reviewer #3: Yes

6. Review Comments to the Author

You may also provide optional suggestions and comments to authors that they might find helpful in planning their study.

Reviewer #1: The authors have addressed all the queries and explained them well. I am satisfied with the answers provided by the authors.

Reviewer #2: Thank you for the feedback given. I am happy that all my concerns have been addressed. Concerns from other authors have also been reviewed and I am happy that their requirements have been met

Reviewer #3: In their article “HIV-associated nephropathy: Protocol and rationale for an exploratory genotypephenotype study in a sub-Saharan African population” the authors describe a planned study by which they try to identify differential expressed gene expression biomarkers in HIV associated nephropathy. The topic is very interesting and there is a lack of available data so far. In their point-by-point response to the earlier reviewer comments they answer in detail all questions and concerns raised sufficiently.

The authors know about the limitation of the study due to the low number of patients which will be included in the study. Due to the high costs for RNAseq it is not possible for this study to include more patients. I like the idea of separating the cohort into two groups (one RNAseq and one validation group) but the authors might consider instead of dividing their patients in 4 groups, to reduce the number of groups to two or three (no, moderate and severe ckd) to get higher patient numbers per group.

7. PLOS authors have the option to publish the peer review history of their article (what does this mean?). If published, this will include your full peer review and any attached files.

Reviewer #1: No

Reviewer #2: No

Reviewer #3: No

---

## [Editor Report · Acceptance letter]

26 Mar 2021

PONE-D-20-21492R1 

HIV-associated nephropathy: Protocol and rationale for an exploratory genotype-phenotype study in a sub-Saharan African population 

Dear Dr. Yusuf:

I'm pleased to inform you that your manuscript has been deemed suitable for publication in PLOS ONE. Congratulations! Your manuscript is now with our production department. 

Kind regards, 

on behalf of

Professor Nicole Endlich 

Academic Editor

PLOS ONE